# Techno-Economic Assessment in a Fluidized Bed Membrane Reactor for Small-Scale H_2_ Production: Effect of Membrane Support Thickness

**DOI:** 10.3390/membranes9090116

**Published:** 2019-09-06

**Authors:** Gioele Di Marcoberardino, Jasper Knijff, Marco Binotti, Fausto Gallucci, Giampaolo Manzolini

**Affiliations:** 1Department of Mechanical and Industrial Engineering, University of Brescia, 25123 Brescia, Italy; 2Inorganic Membranes and Membrane Reactors, Department of Chemical Engineering and Chemistry, Eindhoven University of Technology, 5612 AZ Eindhoven, The Netherlands; 3Politecnico di Milano, Department of Energy, 20156 Milano, Italy

**Keywords:** hydrogen permeation, fluidized membrane reactor, permeance

## Abstract

This paper investigates the influence of the support material and its thickness on the hydrogen flux in Palladium membranes in the presence of sweep gas in fluidized bed membrane reactors. The analysis is performed considering both ceramic and metallic supports with different properties. In general, ceramic supports are cheaper but suffer sealing problems, while metallic ones are more expensive but with much less sealing problems. Firstly, a preliminary analysis is performed to assess the impact of the support in the permeation flux, which shows that the membrane permeance can be halved when the H_2_ diffusion through the support is considered. The most relevant parameter which affects the permeation is the porosity over tortuosity ratio of the porous support. Afterward, the different supports are compared from an economic point of view when applied to a membrane reactor designed for 100 kg/day of hydrogen, using biogas as feedstock. The stainless steel supports have lower impact on the hydrogen permeation so the required membrane surface area is 2.6 m^2^ compared to 3.6 m^2^ of the best ceramic support. This ends up as 5.6 €/kg H_2@20bar_ and 6.6 €/kg H_2@700bar_ for the best stainless steel support, which is 3% lower than the price calculated for the best ceramic support.

## 1. Introduction

The development of innovative technologies based on renewable energy is becoming essential for a more sustainable future. Among alternative energy carriers, hydrogen is certainly the most popular because of the high energy density on mass base and the absence of CO_2_ emissions at the point of use. The global hydrogen production is around 600–720 Billion Nm^3^/year [1], increasing at a rate of 5–6% per year. About 96% of the hydrogen is produced from fossil fuels [2], mostly natural gas (NG) as feedstock accounting for 50% of the world hydrogen production, and only 3–4% is produced without using fossil fuels [3]. The hydrogen produced by fossil fuels cannot be considered CO_2_ neutral, as conventional hydrogen production plants emit large CO2 emissions into the atmosphere (about 380–420 g_CO2_/Nm^3^_H2_) [4]. The production of H_2_ from renewables is usually performed via water electrolysis using the electricity generated by photovoltaic or wind power stations. The resulting price of H_2_ production is around 5 $/kg [2], above the target of 2.4 $/kg (original reference reports the cost in 2 €/kg [5]), and about three times the H_2_ cost from a large-scale steam reforming plant equal to 1.28 $/kg [6]. An alternative option consists of producing the H_2_ from biogas: The conversion process can be similar to the steam reforming one (biogas composition is mainly CH_4_ diluted in CO_2_), but this has to be applied at a much smaller scale. The typical size of existing BG plants corresponds to around 0.20–25 Nm^3^/s of biogas (with a methane content of 40–60%_mol_), which is 100 times smaller than the large-scale reforming plant, but there is a significantly higher number of plants (by the end of 2017, in Europe there were over 17,700 biogas plants for an overall installed capacity of 10.5 MW_el_) [7]. Considering the small-scale of biogas plants, it is mandatory to adopt an efficient system for biogas conversion to hydrogen which is slightly affected by the size of the components. At these scales, the hydrogen price is much higher with conventional systems, as heat recovery is more complicated, and export of steam/electricity is not possible.

Membrane reactors have been always foreseen as a promising technology because of their capability to simultaneously produce and separate the hydrogen [8,9,10,11]. Hydrogen production is performed via steam-reforming reaction and hydrogen separation through H_2_ selective membranes [12]. The problem of integrating membranes in a steam reformer reactor relies in the heat management: The reaction is very endothermic, so heat must be supplied to the reaction side while avoiding damaging the membrane [13]. Fluidized membrane reactors with auto-thermal reforming were demonstrated to solve this problem, as well as reducing the mass transfer limitation of the hydrogen from the bulk of the reaction to the membrane wall [14,15,16].

Among the different types of hydrogen perm-selective membranes, the Pd-based type has always been identified as the most suitable because of the operating temperature range (up to 500–600 °C) similar to the steam-reforming reaction [11,17]. Moreover, they offer the best compromise between permeance, selectivity, and chemical stability [16]. As a drawback, one of the most relevant Pd membrane limitations, which is beyond technological development, is the high cost of palladium [18]. For this reason, Pd membranes can be deposited on a support to reduce the thickness with cost benefits and to increase the mechanical stability with respect to the self-supported one. More resistant membranes can work with larger total pressure differences between the feed and permeate sides, with benefits for the hydrogen permeating across the membrane according to permeation law [19].

Hence, the membrane layer support must have specific properties to allow the deposition of the Pd layer on top of this [20]. One of the most important properties is the pore diameter: On one hand, it shall be significantly lower than the Pd layer, which is between 2 to 5 µm [20]; on the other hand, the smaller the support pores, the higher the resistance to hydrogen permeation and; therefore, the impact of the support to the overall flux [16,21]. An option that partly overcomes this issue is the adoption of asymmetric supports: The substrate where the Pd layer is deposited consists of small pores, while the majority of the support consists of larger pore sizes [18].

Concerning the reactor design, as anticipated, the feed gas pressure has a direct effect on the permeation driving force through the membrane, but also the pressure on the permeate side is relevant. The typical configuration adopted to increase the H_2_ flux consists of decreasing the hydrogen partial pressure in the permeate side by using a vacuum or a sweep gas [22]. The former option leads to simpler plant configurations but lower performance. Using sweep gas would reduce the overall cost as no vacuum pump had to be used on the permeate side [23].

However, some studies suggest that using sweep gas increases the diffusion resistance of the support, thus reducing the driving force across the membrane [24,25,26,27]. Furthermore Zeng et al. and Li et al. discovered that the H_2_ transport out of the ceramic support was slowed down by the counter diffusion of N_2_ [28,29]. According to Pinacci and Drago [22], an increase in sweep gas flow rate decreased the H_2_ flux instead of increasing it, due to the resistance in mass transfer through the support [22]. The main limitation of this work was the absence of the polarization effects in the retentate and the permeate side, as pure hydrogen was used as feedstock to implement the dusty-gas model (DGM). Nordio et al. [18] performed a study on the effect of sweep gas on the supported Pd membranes with concentration polarization and diluted hydrogen. The authors compared three different membranes with different support thickness and with different sweep gas flow. The experimental campaign revealed that the support thickness has a direct effect on the pressure drop, thus on the mass transfer limitation. The mass transfer limitation is inversely proportional to the hydrogen binary diffusivity in the mixture, so higher partial pressure leads to high mass transfer resistance. In addition, a high pressure drop over the porous support gives a negative impact on the interface pressure of the palladium surface increasing the pressure at the interface with penalties on the flux [18]. The experimental results confirmed the importance of accounting for the diffusion barrier of the porous support, so mathematical models have been developed [26,27] or innovative types of membranes as duplex (the Pd layer is on both sides of the tube) have been manufactured [24].

To sum up, the sweep gas benefits can be counterbalanced by the support geometry (thickness) and material properties. Therefore, the identification of the optimal sweep flow requires detailed information on the type of support and its properties and specific modelling studies. In general, ceramic (alumina, α-Al_2_O_3_) and metallic (porous stainless steel (PSS)) materials are used as membrane support. Whilst the first can be easily made to obtain small pores and/or different pore size distribution, but has low mechanical strength, the latter do not suffer the leakage problem thanks to the easier connection to the reactor shell, but shows low hydrogen selectivity due to large pores. Further comparison between the two are given below in Table 1.

Typically, thicker ceramic supports (up to 3.5 mm) are the most common support used in Pd membranes [21,30,31], being cheaper than the metallic ones and ensuring high selectivity.

This works aims at giving more insight on the effect of the support type integrated in a fluidized membrane reactor using sweep gas on the amount of hydrogen permeated at different operating pressures and sweep gas flow rates. A detailed comparison in terms of hydrogen flux and costs between ceramic and porous stainless steel (PSS) supports is performed. The analysis starts with the performance assessment of the reactor over a wide range of operating conditions (feed and permeate pressures, sweep flows) assuming methane as feedstock in the membrane reactor. This analysis points out the support properties that mostly affect the hydrogen permeation. Then, the impact of the support on the hydrogen permeation are applied to biogas as feedstock, so to evaluate the more convenient type of membrane from both thermodynamic and economic point of views. The analysis is carried out using Aspen and integrates the details of the hydrogen permeation phenomena with the overall plant performance. To the authors’ knowledge, this is the first time that this kind of comparison is performed, in particular identifying the most convenient type of support from an economic point of view. This research is performed within the BIONICO European project, where a fluidized membrane reactor capable of producing up to 100 kg/d of H_2_ is manufactured and tested in a real biogas plant site.

The paper is organized as follows: Section 2 describes the mathematical model of the membrane reactor, Section 3 reports the membrane reactor performance as a function of the type of support and its properties. The case study is described in Section 4, presenting the calculated cost of hydrogen for the different considered supports. Finally, the conclusions of the work and future works are outlined in the last section.

## 2. Materials and Methods

The membrane reactor model is developed in Aspen Custom Modeler (ACM) which allows the development of proprietary components to be integrated in Aspen Plus and Aspen Plus Dynamics for steady-state and dynamic simulations of complex chemical/energy systems. The ACM adopted in this work was developed and validated in previous studies performed by Nordio et al. and Foresti et al. [16,18]. The model is based on the dusty-gas model [26,36]. Figure 1 shows the scheme of the fluidized membrane reactor outlining the H_2_ partial pressure trend in presence (left side) or not (right side) of the sweep gas. This model works in a fluidized bed regime, and the bed is divided into three regions. The first region is used for the reaction of steam methane reforming (SMR), water gas shift (WGS), and oxidation (OX). The oxidation zone is considered because an auto-thermal reactor is adopted to simplify the heat management and reactor design [37]. The second region, also known as the membrane region, is used to separate the produced hydrogen. The third region is used as overhead for the expansion of the bed. The three reactions that are happening in the bottom region (SMR, WGS, and OX) are modelled by the kinetic laws from Trimm and Lam, and Numaguchi and Kikuchi [38,39]. The reactor is designed as isothermal, which corresponds to preliminary tests running on the prototype of this reactor [31]. The performance is represented by the hydrogen recovery factor (HRF). As shown in Equation (1), this is defined as the moles of hydrogen permeated over the theoretical recoverable number of hydrogen moles of the complete reforming of the feed (i.e., 4 per each mole of methane), subtracting the methane combusted to support the reaction [16].
(1)$$\mathrm{HRF}=\frac{{\dot{n}}_{H_{2,\mathrm{perm}}}}{4\cdot n_{{\mathrm{CH}}_{4,\mathrm{ret}}}+n_{\mathrm{CO},\mathrm{ret}}+n_{H_{2,\mathrm{ret}}}+n_{H_{2,\mathrm{perm}}}}=\frac{{\dot{n}}_{H_{2,\mathrm{perm}}}}{4\cdot {\dot{n}}_{{\mathrm{CH}}_{4,\mathrm{in}}}-2{\dot{n}}_{O_{2,\mathrm{in}}}}\;.$$

The model can include also the concentration polarization (CP) losses which, as reported in a previous work [16,18], are described by a parameter to be tuned according to the operating conditions (pressure, temperature, compositions). Since for the considered case, this information was not available, the CP losses were neglected; therefore, the partial pressure of H_2_ in the emulsion phase is the same as on the Pd membrane surface, so to only focus on the effect of the membrane support on the flux.

As mentioned in the introduction, the sweep gas plays an important role. The diluent (e.g., N_2_ or steam) is often used as the H_2_ carrier. In this model, steam is used as it is easier to be produced on site and separated. Concerning the support type, the main parameters, which determines the hydrogen permeation, are the porosity, tortuosity, and pore diameter. In Table 2 and Table 3 the characteristics of different types of supports available in literature for both PSS and ceramic are reported. It can be noted that values cover a wide range; therefore, the analysis was limited to selected specific cases.

### Modelling of Pd Membranes:

The permeation is modelled (see Figure 2) assuming a two-step process: Firstly the hydrogen permeates across the Pd membrane, and then the support resistance is considered assuming the sweep gas on the permeate side. The flux through the Pd membrane is described as:(2)$$\begin{array}{c} J_{s}\;\left(\mathrm{mol}/m^{2}s\right)=\frac{\mathrm{Pe}}{t}\left(P_{\mathrm{Feed},H_{2}}^{n}-P_{\mathrm{Perm},H_{2}}^{n}\right)\;. \end{array}$$

Based on experimental data, the H_2_ flux is interpolated and according to Equation (2) the n-value is determined. The membrane permeability is temperature dependent. For this reason, it is described as an Arrhenius temperature dependency Equation (3).
(3)$$\begin{array}{c} \mathrm{Pe}={\mathrm{Pe}}_{o}\exp\left(-\frac{E_{A}}{\mathrm{RT}}\right)\;. \end{array}$$

For the activation energy, the tests reported in [18] are used for the PSS and ceramic supports. The H_2_ diffusion across the Pd membrane is described by Equations (2) and (3), while the DGM [45] (see Equation (4)) is used to account for the diffusion of the H_2_ across the porous support.
(4)$$\overset{n}{\underset{\begin{array}{c} j=1\\ j\neq i \end{array}}{\sum }}\frac{x_{i}N_{j}-x_{j}N_{i}}{{\mathrm{pD}}_{i,j}^{\mathrm{eff}}}-\frac{N_{i}}{{\mathrm{pD}}_{i,K}^{\mathrm{eff}}}=\frac{1}{\mathrm{RT}}\frac{\partial x_{i}}{\partial r}+\frac{x_{i}}{\mathrm{pRT}}\left(\frac{B_{0}p}{{\mu D}_{i,K}^{\mathrm{eff}}}+1\right)\frac{\partial p}{\partial r},$$
where B_0_ is the viscous permeability of the support, D_i,K_ is the Knudsen diffusivity for the i species, and N is the flux of either hydrogen or steam. In this model, the Knudsen diffusion flow (first term), viscous flow (second term), and the binary diffusion (third term) are accounted for [25].

The effective Maxwell–Stefan gas–gas diffusivity [18,46] is given by:(5)$$D_{i,j}^{\mathrm{eff}}=\frac{\varepsilon }{\tau }D_{i,j}\;.$$

The effective Knudsen diffusivity is given by:(6)$$D_{i,K}^{\mathrm{eff}}=\frac{\varepsilon }{\tau }\frac{d_{\mathrm{pore}}}{3}\;\sqrt{\frac{8\mathrm{RT}}{{\pi M}_{i}}}\;.$$

The viscous permeability is a function of the pore diameter, ε and τ. This formula [18,46] is used if convection through the support is modelled as laminar and for cylindrical pores:(7)$$B_{0}=\frac{\varepsilon }{\tau }\frac{d_{\mathrm{pore}}^{2}}{32}\;.$$

## 3. Results

The impact of membrane support on hydrogen permeation is firstly evaluated using natural gas as feedstock. The selection of natural gas is due to the greater availability of experimental measurements and works available in literature to be compared against [16,18,23]. The membrane reactor characteristics and operating conditions are summarized in Table 4. The feed pressure is varied from 8 to 20 bars. Table 4 summarizes the feed streams and reactor conditions for the sweep gas case. Four different cases (see with several support thickness (from 0.5 to 3 mm)) are proposed to fully describe the effect of the pore size and ε/τ and cover a wide range of membrane support types.

The characteristics of the membrane supports investigated are reported Table 5. Figure 2 shows the Pd composite membrane where the three points are, respectively, the emulsion phase, interface, and permeate partial pressure of H_2_. Since, the length of the membrane is kept equal to 0.4 m, the H_2_ flux will decrease along the length.

Figure 3 shows the different pressure gradients for the cases considered with support thicknesses of 0.5 and 3 mm. Even though the initial conditions are the same (limited differences can be noted due to the solid circulation, which is modified by the model depending on the permeated hydrogen), significant differences in the pressure profiles can be noted: Assuming a fixed length, lower hydrogen permeation ends up in higher partial pressure at the feed side. Other considerations are:Case 2 is the one that has the highest penalties in terms of partial pressures with thicker membranes, while Case 4 has very limited differences; Case 2 have roughly the same pore radius of Case 4, while 10 times lower ε/τ;Case 1 and Case 3 have pretty similar with limited impact of the support thickness on the partial pressure trend. Case 1 and Case 3 share similar ε/τ with larger pore diameters for the alumina support (Case 1);

The considerations on the pressures can be translated into the HRF parameter, as shown in Figure 4.
Case 4 shows the highest HRF thanks to the lower resistance introduced by the support, hence a lower driving force requested. Case 1 and Case 3 have almost the same HRF and trend;The support thickness can affect the HRF by more than 20%, ranging from 40% of Case 2 to 60% of Case 4;The permeate pressure is an important factor to justify the higher HRF, with 3 mm supports compared to the one for 2.5 mm supports. As the calculations are performed at constant outer diameter and sweep inlet pressure, the thicker the membrane, the lower the final permeate pressure as a consequence of the higher pressure drops in the permeate side. In the 20 bar case, the permeate pressure can go even below the atmospheric one (this is not a design condition for a real plant, but it serves for the purpose of this analysis).

For sake of completeness, the partial pressure difference for the different cases as a function of the support thickness is reported in the Appendix A.

A further analysis is carried out to elucidate the different contributions of the pore diameter and ε/τ on the hydrogen permeation (see Figure 5) assuming a feed pressure of 15 bar and pore diameter of 0.36 µm. The HRF is affected by any ε/τ value for thicker supports (3 mm), while only for the very low values (below 0.4) for thicknesses of 0.5 mm. Moving to the pore size impact, a similar trend to the ε/τ parameter can be pointed out. The main difference is in the magnitude of the impact: The ε/τ can change the HRF from 0.35 to 0.55, while the pore size has half the impact as the HRF ranges from 0.32 to 0.46. To sum up, a higher ratio of ε/τ will diminish the effect of the support, since less collisions with the wall occurs during H_2_ permeation and a higher Knudsen diffusion occurs. For small pores, the thickness of the support becomes more important. Increasing the pore size, the limitations are minimized. Hence, a turning point occurs in the HRF because less collisions to the wall happens.

As anticipated, the effect of varied sweep flow also plays an important role in maximum-achieved HRF according to the two studies [18,22,27].The calculations performed in this work at varying sweep flow rates are reported in Figure 6, using the two thickness’ based on mechanical stability and sealing issues equal to 0.5 and 2 mm [31] for 316 L and α-Al_2_O_3_, respectively. On the x-axis, a sweep flow to H_2_ permeated ratio is used because it allows implementation of the results in different models. A clear asymptotic trend can be seen confirming the limited advantages of using large sweep flow rates. This graph also shows the HRF when diffusion effects are neglected, whereby the stainless steel Case 4 (i.e., high ε/τ) comes close to the ideal case.

This preliminary analysis confirmed the importance of accounting for the diffusion limitation in the support, as it reduces the hydrogen permeated. In general, the metallic and the ceramic supports show similar performance when they have similar thicknesses and ε/τ. It seems that metallic membranes can have higher ε/τ, with benefits for the hydrogen permeation. Certainly, the thickness of the ceramic support shall be higher than the metallic ones to limit the leakages in correspondence to the sealings.

Finally, a new parameter called equivalent permeance is introduced to quantify the impact of the support in the permeation process. The equivalent permeance represents the value of the Pd membrane permeance without any support, which will determine the same amount of hydrogen permeated of the actual Pd membrane with the support. It is determined changing the permeance value in the model so as to achieve the same results above reported.

The values of the equivalent permeance determined for the different cases as a function of the support thickness are reported in Figure 7. The effect of the support can even half the initial permeance value of the Pd membrane in the case of thick support and high ε/τ.

In the next section, a case study will be introduced so as to evaluate both the thermodynamic performance and the economic costs of the different supports considered.

## 4. Case Study

The different supports are integrated in a membrane reactor for hydrogen production in the BIONICO reactor. The BIONICO reactor is designed to produce 100 kg/day of pure hydrogen, starting from a landfill type of biogas, with the following molar composition: CH_4_ equal to 44.2%, CO_2_ equal to 34.0%, N_2_ equal to 16.0%, and O_2_ equal to 2.7%. The plant layout (Figure 8), which was introduced in a previous work [21], is here briefly discussed. Table 6 summarizes the thermodynamic properties of the main streams involved for one of the simulation case with landfill biogas [21]. The biogas is compressed, then preheated up to 300 °C in HX-3. It is mixed with steam and water before final heating in HX-4 prior to the inlet of the ATR-MR. At the ATR-MR outlet, the retentate (point 3) and hydrogen (point 6) leave from the top section of the reactor. The retentate stream, mainly consisting of H_2_O, CO_2_, and N_2_ with the remaining fraction of CO, CH_4_, and H_2_ is firstly cooled in a series of heat exchangers down to 200 °C (point 4) and then sent to a catalytic burner for a complete combustion of the remaining fuels with an additional air stream. The thermal power released during retentate, exhaust gases, and permeate cooling covers the heat duty for the steam production (steam for the sweep gas and for the reforming reaction). In particular, the sweep gas steam is evaporated, cooling the exhaust gas and superheated using the permeate stream. The geometry adopted for the heat transfer corresponds to counter-current configuration with hot gas flow externally to staggered tube bundles and cold streams in tube. Such configuration provides compact design while keeping a high heat transfer coefficient. The resulting exchange area is then used for the economic analysis. The separators are simulated as simple adiabatic flash units. The main parameters values adopted for the system auxiliary components (compressors, fans, pumps, heat exchangers, etc.) are taken from previous works [20].

The plant layout is implemented in Aspen Plus^®^ [47], where mass and energy balances are solved. The methodology adopted is consistent with previous works [16,21,48]. The Peng–Robinson cubic equation of state [49] is used for all thermodynamic properties, except for liquid molar volume evaluation where the Rackett model [50] is used, and for steam properties where NBS/NRC steam tables [51] are adopted. To save computational time, the achievements of the previous analysis (in particular the equivalent permeance) are included in the model implemented. The number of membranes and, consequently, the reactor design are determined so as to achieve the HRF that closes the thermal balance of the system avoiding any additional biogas input.

The membrane reactor characteristics are reported in Table 7, pointing out the significant area variation (i.e., number of tubes) between the different cases requested for producing the same amount of hydrogen. As a term of reference, the area calculated in a previous work for the BIONICO project with simplified calculation of the support diffusion resistance is also added.

## 5. Economic Analysis and Discussion

Starting from the results determined in the previous section, an economic analysis is carried out to identify the most cost effective configuration between the different reactor designs as a function of the support type. This analysis is important as stainless steel supports are much more expensive than ceramic ones [52]. This analysis has an accuracy of ±50% [53] because of the development stage of the components. However, it will provide a good indication on the most suitable configuration to be adopted.

The parameter of merit adopted for comparing the different cases is the cost of hydrogen production (LCOH) [2], defined as:(8)$$\mathrm{LCOH}=\frac{\left(\mathrm{TPC}*\mathrm{CCF}\right)+C_{O\&M_{\mathrm{fix}}}+\left(C_{O\&M_{\mathrm{var}}}*h_{\mathrm{eq}}\right)}{{\mathrm{kg}}_{H_{2}}}\;,$$
where TOC is the total overnight cost, CCF is the capital charge factor, $C_{O\&M_{\mathrm{fix}}}$ is the fix operating and maintenance (O&M) costs, while $C_{O\&m_{\mathrm{var}}}$ is the O&M variable costs. The fixed cost includes insurance, maintenance, and labor wages, while the variable cost accounts for water, fuel, and the cost of electricity consumed. The equivalent operating hours is equal to 7500 h and the CCF is equal to 0.13 (corresponding to a plant lifetime of 15 years was considered with discount rate of 10%), consistent with [48,54].

Table 8 summarizes the methodology to calculate the TOC. The equipment cost is determined as in Equation (9).
(9)$$C_{i,2018}={\left(C_{i,0}{\left(\frac{S_{i}}{S_{i,0}}\right)}^{f}\right)}_{y}\times \frac{{\mathrm{CEPCI}}_{2018}}{{\mathrm{CEPCI}}_{y}}\;.$$

The component cost actualized to 2018 is scaled and actualized by this formula to account for the different sizes (S_i_) with respect to the reference size (S_i,0_) and cost (C_i,0_) by a scale factor f. Moreover, to actualize the component cost the chemical engineering plant cost index (CEPCI) of the year 2018 is used, where this value is equal to 603.1 [55]. The corresponding CEPCI of the reference equipment cost should be applied.

The bare equipment costs for the different plant components is summarized in, they are taken from [48] (Table 9).

Table 10 reports the O&M fixed and variable costs (i.e., catalyst, biogas, water, auxiliaries, maintenance, insurance, and operators costs). For the cost of ceramic and stainless steel, data analysis carried out in Ferret project is used [52]. This value includes the cost of the support, membrane manufacturing, and the cost of the selective membrane layer. The used catalyst is the BIONICO catalyst, whose quantity is defined by the ACM model (50% catalyst and 50% filler particles).

Table 11 summarizes the CAPEX and OPEX for the different cases considered. As expected, Case 2 has the highest cost of the membrane reactor as a consequence of the larger membrane area/tubes (+60%), though the costs increase is quite limited (+15%). Case 4 has the lowest reactor cost because of the smaller membrane surface area; however, Case 1 has the cheapest membrane because of the lower price of the ceramic support over stainless steel. From an overall point of view and considering the cost assessment accuracy, Case 1, Case 2, and Case 4 share the same TOC.

The fixed OPEX is mainly determined as a fixed percentage of the TOC; therefore, there is a proportionality between the two costs and no significant difference between Cases 1, 3, and 4 can be pointed out (as it was for CAPEX).

The variable cost (the cost of process water, biogas, and electricity) remains the same for all cases because they all have the same HRF and efficiency. However, the membrane replacement cost and the catalyst is very significant with impacts on O&M_var_. The cost reduction is mainly in the amount of catalyst needed, proportional to the size of the reactor, while limited differences can be pointed out for the membrane replacement costs (as it was in the CAPEX, because the cheaper support balances the higher membrane area). Variable O&M indicates a limited preference for the stainless steel support with respect to the ceramic one

Finally, the LCOH is reported in Figure 9 considering a total amount of H_2_ produced per year equal to 31,250 kg/y. Case 4 has the lowest LCOH, equal to 5.3 €/kg H_2@20bar_ and 6.3 €/kg H_2@700bar_. Cases 1 and 3 have the same LCOH in the range of 5.4 €/kg H_2@20bar_ and 6.4 €/kg H_2@700bar_, so with differences below 3% with respect to the optimal case. Case 4 has a LCOH higher by 10% with respect to Case 1. The accuracy of the economic assessment does not allow the drawing of any final conclusions, but certainly the higher price of stainless steel tubes can be balanced by the higher permeability leading to the competitive cost of LCOH.

Considering the limited accuracy of the economic analysis a sensitivity analysis on the main cost assumption was performed and the main outcomes reported in Figure 10. Results show that the overall capital cost and electricity price have the highest impact on LCOH leading to a variation around 10%. The other parameters are less relevant and, in particular, reducing the cost of the membrane by 50% results in only a 5% LCOH cost reduction.

The calculate LCOH is significantly higher than the target H_2_ price, which is 2 €/kg [5]. The higher price is due to the smaller-scale production (which does not allow export of steam/electricity), high cost of biogas, and the need for further development in the membrane technology focusing on higher permeability, lower manufacturing costs, and increased lifetime. However, the sensitivity analysis pointed out that the considered layout is penalized because of the small-scale and the utilization of biogas as feedstock, with respect to the adoption of natural gas.

## 6. Conclusions

This paper performed a detailed techno-economic assessment on support thickness for hydrogen production in a membrane reactor, using Palladium membranes in fluidized bed conditions in the presence of sweep gas. The advantage of stainless steel is the easy integration, but the disadvantage if the high cost. Ceramic supports are cheaper but need to be thicker since integration is more difficult at higher temperatures. The effect of the support used has been investigated in detail to identify the most relevant characteristics. It was found that the porosity over tortuosity ratio is more relevant than the pore diameter, as it significantly affects gas diffusion. To assess the impact of the support on the permeance, an equivalent permeance is introduced. For the best case scenario, stainless steel has a reduction of 5% with respect to a pure Pd membrane. For the ceramic cases, this parameter shows a reduction ranging between 30% and 60% compared to the reference condition.

These results are applied to a fluidized membrane reactor operating with biogas for hydrogen production equal to 100 kg/day. This reactor is currently under testing in the BIONICO project (summer 2019). The analysis showed that, for the same biogas and hydrogen production, the best stainless steel support requires 2.6 m^2^ compared to 3.6 m^2^ of the best ceramic support, meaning that the reactor is also larger. The resulting cost of hydrogen production resulted equal to 5.6 €/kg H_2@20bar_ and 6.6 €/kg H_2@700bar_ for the best stainless steel support, which is 3% lower than the price calculated for the best ceramic support. The main difference is related to the variable O&M costs as a consequence of the larger membrane reactor. The accuracy of the economic assessment does not allow for the drawing of any final conclusions, but certainly the higher price of stainless steel tubes can be balanced by the higher permeability leading to the competitive cost of LCOH.

## Figures and Tables

**Figure 1 membranes-09-00116-f001:**
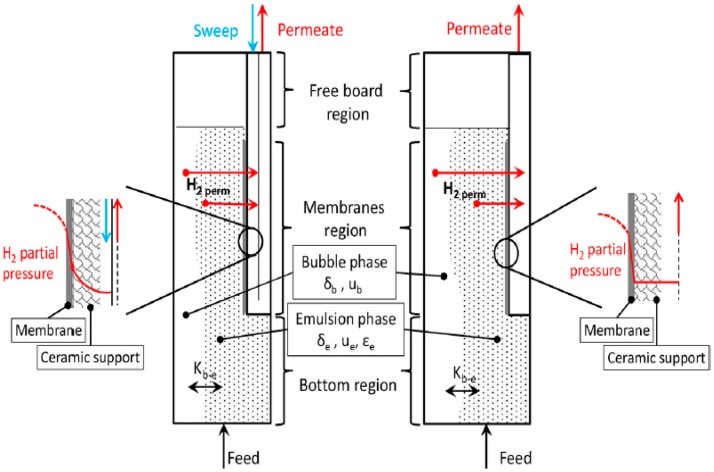
Scheme of the fluidized bed membrane reactor developed in ACM with and without gas for PSS and ceramic supports [16].

**Figure 2 membranes-09-00116-f002:**
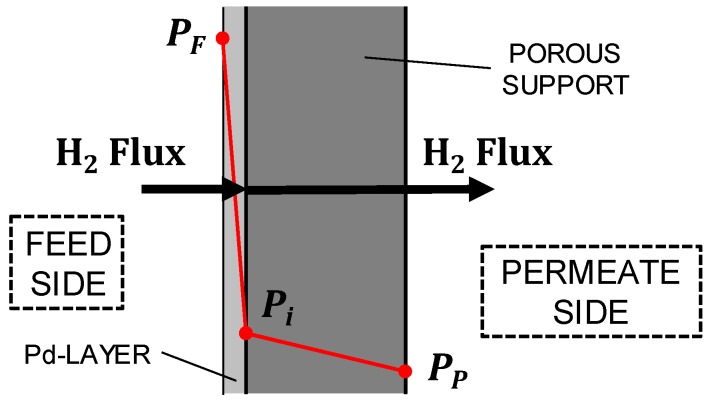
Model of the Pd composite membrane [22].

**Figure 3 membranes-09-00116-f003:**
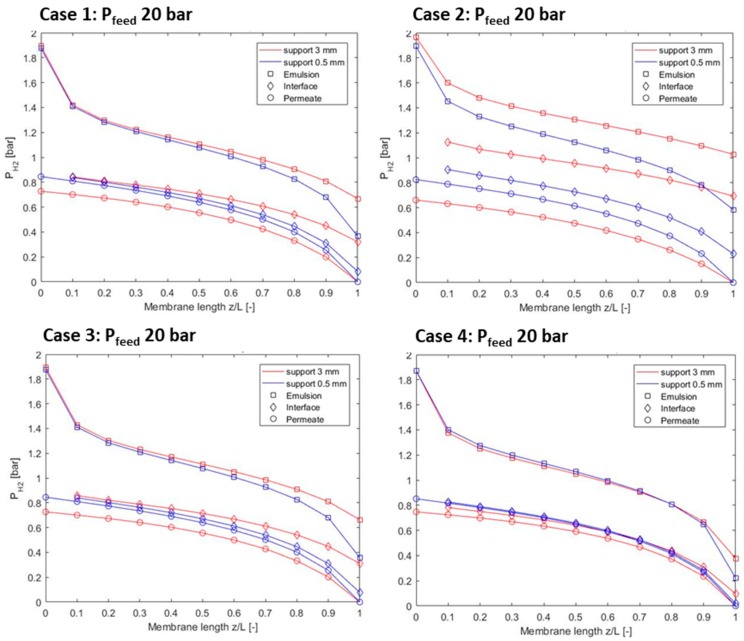
Pressure gradients comparison for the considered cases.

**Figure 4 membranes-09-00116-f004:**
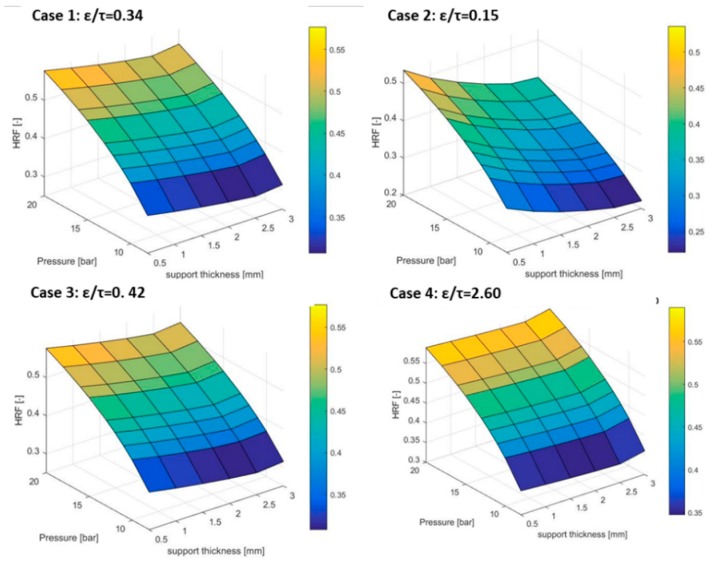
HRF comparison for the considered cases.

**Figure 5 membranes-09-00116-f005:**
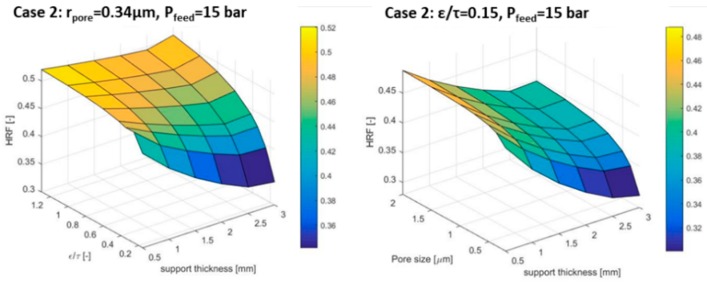
Porosity over tortuosity (left side) and pore size (right side) effects on HRF.

**Figure 6 membranes-09-00116-f006:**
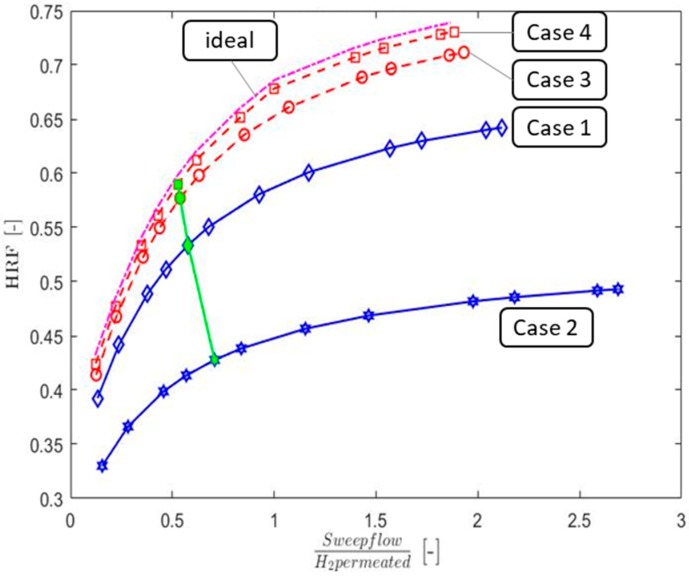
Sweep flow rate impact on the HRF for the considered cases.

**Figure 7 membranes-09-00116-f007:**
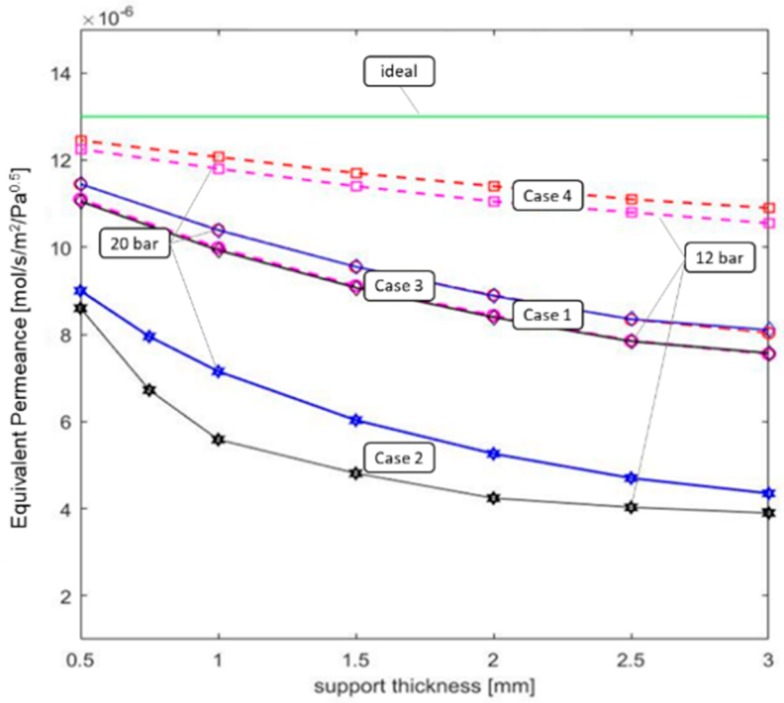
Equivalent permeance calculated at 12 and 20 bar feed pressure for different support thicknesses.

**Figure 8 membranes-09-00116-f008:**
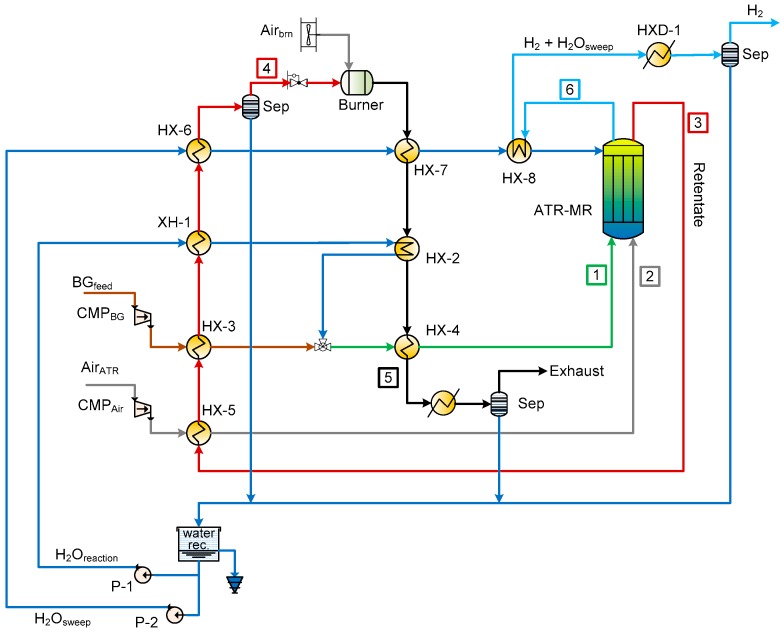
Layout of BIONICO system using sweep gas.

**Figure 9 membranes-09-00116-f009:**
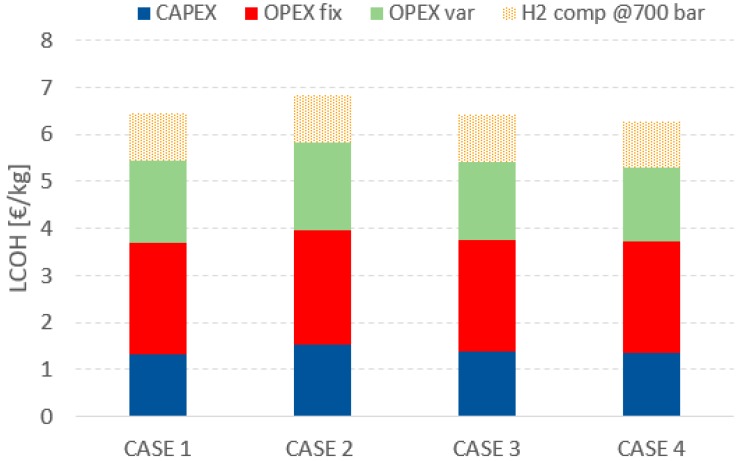
LCOH comparison for the considered supports.

**Figure 10 membranes-09-00116-f010:**
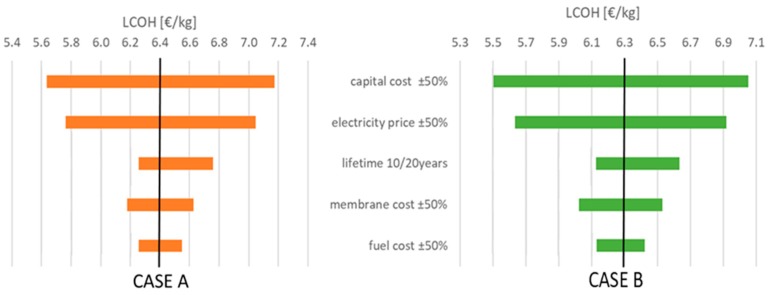
Sensitivity analysis on main economic assumptions for Case A (left) and Case B (right). LCOH cost calculated assuming H_2_ at 700 bar.

**Table 1 membranes-09-00116-t001:** Difference between ceramic and porous metallic supports [32,33,34,35].

Support Material	Advantages	Disadvantages
Ceramic supports	Variety of materials Pore structures Qualities	Brittle Integration is difficult at high temperature
Metallic supports	Easy integration Mechanical stability Same thermal coefficient	Large open pores Wide size distribution High surface roughness

**Table 2 membranes-09-00116-t002:** Porous metallic support data.

Pore Size	Porosity/Tortuosity	Support Geometry	Surface Area	Manufacturer	Reference
0.5 μm	-	Thickness 1 mm	7.07 cm^2^	Mott metallurgical corporation	[40]
2 μm	-	Thickness 0.48 mm	6.8 cm^2^	AccuSep PALL corporation USA	[41]
0.2 μm	ε = 20–23%	OD 6.4 mm ID 3.2 mm	9 cm^2^	Mott Metallurgical Corporation	[42]
3 μm	ε = 17%	OD 15.9 and 12.7 mm Thickness 1.6 mm	-	Mott Metallurgical Corporation	[43]
0.5 μm 1–2 μm	-	OD 10 mmID 6 mm	-	GKN Sinter Metal Filters GmbH Radevormwald	[44]
0.45 μm	ε/τ = 1.12	OD 21 mm Thickness 1 mm	-	Mott Metallurgical Corporation	[32]
0.30 μm	ε/τ = 2.89				
0.87 μm	ε/τ = 0.42				
0.24 μm	ε/τ = 2.60				

**Table 3 membranes-09-00116-t003:** Ceramic support data.

Pore Size	Porosity	Tortuosity	Thickness	Surface Area	Manufacturer	Reference
0.16 μm	ε = 35%	τ = 1.25	14 mm OD 10 mm ID	21 cm^2^	Provided by ECN	[27]
0.16 μm	ε = 35%	τ = 1.25	14 mm OD 10 mm ID	21 cm^2^	Provided by ECN	[27]
3.4 μm	ε = 43%	τ = 1.25	14 mm OD 10 mm ID	21 cm^2^	Provided by ECN	[27]
0.36 μm	ε/τ = 0.15		1.5 mm	29/36.4 cm^2^	-	[16]

**Table 4 membranes-09-00116-t004:** Feed stream and reactor conditions, membrane properties [16,27,32].

Feed Stream Conditions	Value	Membrane Characteristics	Value
Natural gas feed (kmol/h)	0.055	Membrane thickness (μm)	5
Steam feed (kmol/h)	0.165	Permeance (kmol s^−1^ m^−2^ Pa^−n^)	1.3 × 10^−5^
Air feed (kmol/h)	0.062	Ea (J/mol)	10,171
Sweep gas (kmol/h)	0.058	n	0.5
Temperature (°C)	400	Membrane length (m)	0.4
Pressure (bar)	8–20	Outer diameter support (mm)	10
S/C ratio	3	Inner diameter support range (mm)	4–9
		Membrane area (m^2^)	0.14
**Reactor conditions**			
Temperature (°C)	500		
Pressure (vessel/permeate side) (bar)	8–20/1.3		

**Table 5 membranes-09-00116-t005:** Characteristics of the different cases investigated in this work.

Cases	Support Type	OD [mm]	ID Range [mm]	ε/τ	r_p_ [μm]
**Case 1** [27]	α-Al_2_O_3_	10	4–9	0.344	3.4
**Case 2** [16]	α-Al_2_O_3_	10	4–9	0.15	0.34
**Case 3** [32]	316 L	10	4–9	0.42	0.87
**Case 4** [32]	316 L	10	4–9	2.6	0.24

**Table 6 membranes-09-00116-t006:** Stream properties at in/exit of reactors (@pressure and T).

Stream	Flow		T (°C)	p (bar)	Composition (% Molar Basis)						
	Molar (mol/s)	Mass (g/s)			CH_4_	H_2_	CO	CO_2_	H_2_O	O_2_	N_2_
1	1.16	27.35	535	20	24.9	-	-	19.2	45.4	1.5	9.0
2	0.32	9.32	520	20	-	-	-	-	-	21	79
3	1.24	35.51	550	20	6.4	5.4	2.0	32.9	24.3	-	29.1
4	0.9	28.29	30.1	20	8.6	7.4	2.7	41.1	0.2	-	39.9
5	2.32	70.76	335	1.1	-	-	-	20.2	9.6	4.7	65.5
6	1.15	11.56	550	1.1	-	50.0	-	-	50.0	-	-

**Table 7 membranes-09-00116-t007:** New reactor dimensioning compared to BIONICO.

Parameters	Case 1: (r_pore_ = 3.4 μm) ε/τ = 0.344	Case 2: (r_pore_ = 0.36 μm) ε/τ = 0.15	Case 3: (r_pore_ = 0.87 μm) ε/τ = 0.42	Case 4: (r_pore_ = 0.24 μm) ε/τ = 2.60
**Thickness (mm)**	2	2	0.5	0.5
**N_mem_ (–)**	267	438	223	203
**A_mem_ (m^2^)**	3.36	5.50	2.80	2.55
**V_vessel_ (m^3^)**	0.25	0.41	0.22	0.20
**m_cat_ (kg)**	165	175	135	120
**HRF (%)**	62.40	62.80	62.40	62.50

**Table 8 membranes-09-00116-t008:** Methodology for the TOC calculation [56].

**Plant Component**	**Cost (M €)**
Compressor	A
Heat exchanger	B
Reactor	C
**Bare erected cost (BEC)**	**A + B + C**
Direct costs as percentage of BEC Includes piping/valves, civil works, instrumentation, steel structure, erections, etc.	
Total installation cost (TIC)	80% BEC
**Total direct plant cost (TDPC)**	**BEC + TIC**
Indirect costs (IC)	14% TDPC
**Engineering procurement and construction (EPC)**	**TDPC + IC**
Contingencies and owner’s costs (C&OC)	
Contingency	5% EPC
Owner’s cost	10% EPC
Total C&OC	15% EPC
**Total plant cost (TPC)**	**EPC + C&OC**

**Table 9 membranes-09-00116-t009:** Cost assumption for plant components.

Components	Amount	Scaling Parameter	S_0_	C_0_ (k€)	f	Year Cost	CEPCI
**Reactor**	1	Weight (lb)	130,000	70.32	0.3	2007	525.4
**Heat Exchanger**	8	Exchange area (m^2^)	2	15.5	0.59	2007	525.4
**Biogas compressor**	1	Power (kW)	5	3.3	0.82	2006	499.6
**Air compressor**	1	Power (MW)	0.68	3.42	0.67	2009	521.9
**Water demineralizer**	1	Water flow rate (l_H2O_/h)	90	2.1	0.68	2011	585.7
**Water pump**	2	Water flow rate (l_H2O_/h)	90	1.2	0.7	2011	585.7

**Table 10 membranes-09-00116-t010:** Assumptions to calculate the O&M costs [48,52,56].

**O&M—Fixed**	
Labor costs	60,000 €
Maintenance costs	2.5% TOC
Insurance	2.0% TOC
**O&M—Variable**	
Catalyst cost	258 €/kg/y
Filler particles	12 €/kg/y
Membrane replacement ceramic	360 €/m^2^/y
Membrane replacement SS	2040 €/m^2^/y
Deionization Resin	90 €/y
Lifetime	5 Years
Process water	0.35 €/m^3^
Biogas cost	1.50 €/GJ_LHV_
Electricity cost	0.12 €/kWh

**Table 11 membranes-09-00116-t011:** CAPEX and OPEX for the different cases considered.

Components	Ceramic		Stainless Steel	
	Case 1: (r_pore_ = 3.4 μm) ε/τ = 0.34	Case 2: (r_pore_ = 0.36 μm) ε/τ = 0.15	Case 3: (r_pore_ = 0.87 μm) ε/τ = 0.42	Case 4: (r_pore_ = 0.24 μm) ε/τ = 2.60
**CAPEX**				
Reactor cost (k€)	29.5	34.2	28.4	27.6
Membranes (k€)	22.8	37.4	28.6	26.0
Heat exchangers (k€)	73.9	73.9	73.9	73.9
Biogas compressors (k€)	3.9	3.9	3.9	3.9
Balance of plant (k€)	0.3	0.3	0.3	0.3
H_2_ compr @20 bar (k€)	4.4	4.4	4.4	4.4
**TPC @20 bar (k€)**	**318.2**	**363.7**	**329.0**	**321.1**
H_2_ compr @700 bar (k€)	22.1	22.1	22.1	22.1
**TPC @700 bar (k€)**	**370.3**	**415.8**	**381.2**	**373.3**
**OPEX**				
Catalyst + filler	22.2	23.6	18.2	16.2
Biogas	9.1	9.1	9.1	9.1
Water cost	0.2	0.2	0.2	0.2
Electricity @20 bar	18.5	18.5	18.5	18.5
Membranes	4.6	7.5	5.7	5.2
Deionization resin	0.45	0.45	0.45	0.45
**O&M_var_. total @ 20 bar**	**55.0**	**59.3**	**52.1**	**49.6**
Electric energy @ 700 bar	21.6	21.6	21.6	21.6
**O&M_var_. total @ 700 bar**	**76.7**	**81.0**	**73.8**	**71.3**
**O&M_fix_ @ 20 bar**	**74.32**	**76.36**	**74.81**	**74.45**
**O&M_fix_ @ 700 bar**	**76.66**	**78.71**	**77.15**	**76.80**

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
