# Peer review of "Techno-Economic Assessment in a Fluidized Bed Membrane Reactor for Small-Scale H2 Production: Effect of Membrane Support Thickness"

_membranes, 2019, doi:10.3390/membranes9090116_

Round 1
Reviewer 1 Report
The projection price (target price) of hydrogen is 2$/kg but here we can see other values such as 3$/kg and 5$/kg. Comment on this. Table 1 needs further information such as thermal expansion, strength, creep, yield, pore size. A discussion on elemental analysis of the ceramic support can clarify the type of the ceramic. Qualitative comparison is not scientific. Authors could do the comparison in the text and remove the table. More elaboration on Fig. 1 is required. The assumption list and also the method of solution must be further discussed. Which software was employed to solve the equations? (7)-(9) need proper support or citations from the literature. Validation of the model against the experimental data is required to show the reliability of the model. Fig 8: The layout of the system must be further described. Type of the HEX, burner, separator and other units should be discussed. For example, for separator, is it just a thermodynamic separator or there is a specific configuration considered for it? What is the amortising period for the assessment of the process? Table 9: What is the reference case for the cost assumption given in Table 9? Provide the citation. A sensitivity analysis is required between LCOH and the cost assumptions used in Table 9 .Author Response
Please see the attachment

Reviewer 2 Report
Techno-economic assessment on support thickness 3 for Pd-membrane in fluidized bed reactor for small 4 scale H2 production review
The manuscript presents an interesting topic in the techno-economic assessment in different membranes, however, in my opinion, minor corrections in the manuscript should be done before considering it for publication.
I suggest that the authors get editing help from someone with full professional proficiency in English to revise the entire English writing through the manuscript The novelty needs to be highlighted in the introduction as I struggled to find the novelty In your introduction section, you mentioned “Hydrogen has a higher energy density” which is full picture as the energy density of hydrogen compressed at 700 bar is 9.17 MJ/L, while it is 35.8 MJ/L for diesel at atmospheric pressure, so in term of safety and energy density, hydrogen fuel is in favourable, please comment it in the manuscript. In your introduction section, you mentioned the methane in biogas applications, however, you also need to mention the methane slip challenge as well.
Round 2
Reviewer 1 Report
Authors have addressed my comments. Paper can be accepted now.